# FedChill: Adaptive Temperature Scaling for Federated Learning in Heterogeneous Client Environments

## Abstract

Federated Learning (FL) enables collaborative model training with data privacy but suffers in non-i.i.d. settings due to client drift, which degrades both global and local generalizability. Recent works show that clients can benefit from lower softmax temperatures for optimal local training. However, existing methods apply a uniform value across all participants, which may lead to suboptimal convergence and reduced generalization in non-i.i.d. client settings. We propose **FedChill**, a heterogeneity-aware strategy that adapts temperatures to each client. FedChill initializes temperatures using a heterogeneity score, quantifying local divergence from the global distribution, without exposing private data, and applies performance-aware decay to adjust temperatures dynamically during training. This enables more effective optimization under heterogeneous data while preserving training stability. Experiments on CIFAR-10, CIFAR-100, and SVHN show that FedChill consistently outperforms baselines, achieving up to 8.35% higher global accuracy on CIFAR-100 with 50 clients, while using $2.26\times$ fewer parameters than state-of-the-art methods.

## 1 Introduction

Federated Learning (FL) enables collaborative model training across decentralized edge devices while preserving privacy, since clients share only model updates rather than raw data (McMahan et al., 2017). In practice, however, heterogeneous (non-i.i.d.) client data induces client drift and weight divergence, which significantly degrade both global performance and personalization (Yan et al., 2023; Li et al., 2019; Lee et al., 2024). To mitigate these challenges, numerous methods have been proposed. FedAvg (McMahan et al., 2017) provides the foundation but struggles under heterogeneity (Li et al., 2019). Subsequent extensions include FedProx (Li et al., 2020) with a proximal term, Moon (Li et al., 2021) with contrastive learning, FedProto (Tan et al., 2021) with prototype aggregation, FedGen (Zhu et al., 2021) using generative data sharing, and FedAlign (Mendieta et al., 2022) for feature alignment. In parallel, Knowledge Distillation (KD) (Hinton et al., 2014) has been integrated into FL, giving rise to Federated KD methods that reduce communication and mitigate heterogeneity (Wu et al., 2022; Li et al., 2024), such as FedMD (Li & Wang, 2019) for mutual distillation and FedHKD (Chen et al., 2023) for data-free hyper-knowledge distillation.

Despite these advances, important limitations persist. Regularization- and contrastive-based methods (e.g., FedProx, Moon) only partially address knowledge transfer (Li et al., 2020; 2021). Prototype-based approaches like FedProto require carefully chosen representatives and scale poorly with diverse clients (Tan et al., 2021), while generative methods such as FedGen impose computational overhead and introduce privacy concerns through synthetic data (Zhu et al., 2021; Chen et al., 2023). As a result, effectively addressing statistical heterogeneity in federated learning remains an open challenge.

To address this, logit chilling (Lee et al., 2024) shows that fractional temperature values can mitigate client data heterogeneity during local training. While low temperatures (e.g., $T = 0.05$) may accelerate convergence by sharpening gradients, they can also introduce instability and overfitting, especially in high-capacity models or in scenarios with varying client heterogeneity (Lee et al., 2024). Furthermore, uniform temperature scaling across clients overlooks heterogeneity differences,

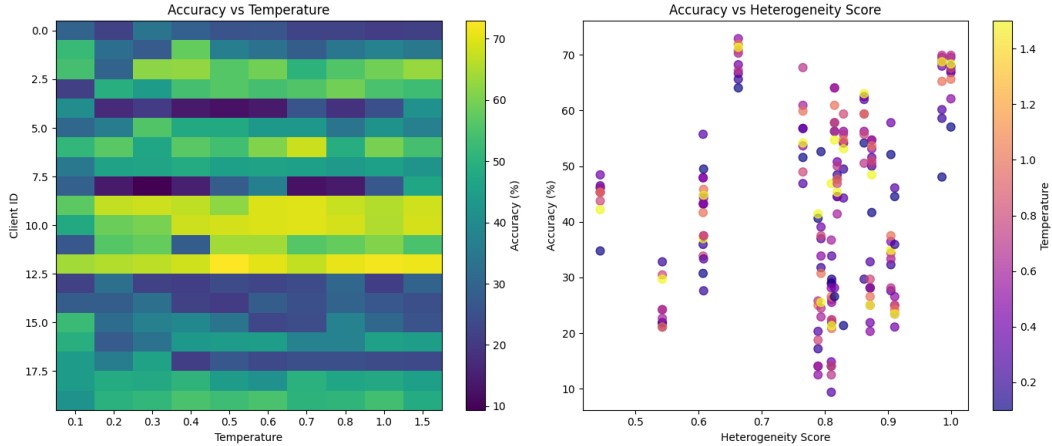

Figure 1: Validation of the temperature-heterogeneity hypothesis. (Left) Client-wise heatmap of validation accuracy across temperatures, showing varying optimal temperatures. (Right) Scatter plot of client heterogeneity vs. accuracy, colored by temperature, highlighting performance trends

making a static one-size-fits-all approach ineffective in non-i.i.d. settings (Lee et al., 2024). These limitations highlight the need for a more comprehensive solution that can effectively address client drift without requiring public data or imposing significant computational overhead.

To this end, we propose **FedChill**, a dynamic, context-aware temperature strategy that adapts to each client's needs and training stage. FedChill introduces adaptive temperature initialization as well as temperature chilling for each client during training. The contributions of our proposed FedChill framework are as follows:

1. We introduce a novel method to compute client-specific heterogeneity scores by comparing each client's local class distribution to a globally approximated distribution, constructed without sharing private data. This score quantifies the divergence and drives the personalized temperature initialization process.

2. Based on each client's heterogeneity score, we introduce a **per-client exponential temperature initialization strategy** that ensures each client starts with a temperature value uniquely suited to the distribution and complexity of its local data.

3. During training, FedChill adopts a **unique adaptive temperature decay mechanism** that decreases temperature by a factor once a tolerance parameter is triggered (based on the clients local accuracies) to cater to enhanced client personalization through sharpened gradient signals, as well as improved server performance.

## 2 PROBLEM FORMULATION

Recent research shows that applying **lower temperatures** ($T \in (0,1)$) during the training process can improve convergence and accuracy, especially in heterogeneous federated learning scenarios (Lee et al., 2024). For example, consider the softmax operation (Hinton et al., 2014; Guo et al., 2017) with a temperature parameter $T$, which transforms logits $z_i$ into class probabilities $p_i$ as follows:

$$p_i = \frac{\exp(z_i/T)}{\sum_j \exp(z_j/T)} \tag{1}$$

When $T < 1$, the exponential effect in equation 1 is amplified, leading to a sharper probability distribution where the model is more confident in its predictions. From a training perspective, the gradient of the cross-entropy loss with respect to the logit $z_i$ under temperature-scaled softmax can be expressed as:

$$\frac{\partial \mathcal{L}}{\partial z_i} = \frac{1}{T}(p_i - y_i) \tag{2}$$

where $y_i$ is the ground truth label. It shows that the gradient magnitude is inversely proportional to the temperature, indicating that lower temperatures increase the sensitivity of the loss w.r.t the logits.

## 2.1 COMPLEXITY OF THE TEMPERATURE SEARCH SPACE

As illustrated in Figure 1, the relationship between temperature and client performance reveals a highly complex and client-specific optimization landscape. The left panel demonstrates that different clients achieve optimal performance at vastly different temperature values, with some clients performing best at $T = 0.1$ while others require $T = 0.5$ or higher. This client-wise variation in optimal temperatures creates a challenging optimization problem: no single universal temperature can capture the heterogeneous requirements of all clients, and even small changes in temperature can yield large performance shifts, especially for clients with skewed or highly diverse data.

## 2.2 HETEROGENEITY AS A PREDICTIVE HEURISTIC

The right panel of Figure 1 provides a crucial insight: each point represents a client, with the x-axis indicating the client's heterogeneity score (divergence from global distribution) and the y-axis showing validation accuracy. Points are colored by the temperature value that achieved that accuracy. The visualization demonstrates that different clients achieve their best performance at vastly different temperature values. Crucially, there exists a non-uniform relationship: clients with higher heterogeneity scores tend to benefit from lower temperatures to sharpen their predictions, whereas representative clients align with higher values. This confirms that no single universal temperature can effectively serve all clients.

Our empirical analysis across multiple heterogeneity measures in Appendix A.5 further supports this hypothesis. While individual measures show varying correlation strengths with optimal temperatures, the **heterogeneity score** demonstrates the most consistent negative correlation (e.g., $r = -0.389$), suggesting that clients with higher heterogeneity benefit from lower temperatures. This relationship provides a principled foundation for temperature initialization rather than random or uniform selection.

## 2.3 FORMAL PROBLEM STATEMENT

The fundamental limitation of existing temperature-scaling approaches is that $T$ remains fixed during training across all the nodes $n \in \{1, \cdots, N\}$, despite the demonstrated need for client-specific optimization. To overcome this, we formulate two primary objectives. The first is to devise a heterogeneity score that can predict a unique initial temperature $T_{n,0}$ for each client $n$, leveraging the observed correlation between data distribution divergence and optimal temperature regimes. The second goal is to develop a strategy to identify stagnation during communication round $k$ and adaptively decay $T_{n,k}$ such that it dynamically balances both model performance and training stability.

This leads to the following client-specific and round-adaptive softmax formulation:

$$p_{i,n,k} = \frac{\exp(z_{i,n,k}/T_{n,k})}{\sum_j \exp(z_{j,n,k}/T_{n,k})}. \tag{3}$$

and the corresponding modified cross-entropy loss becomes:

$$L = -\sum_{i=1}^{N} y_i \log(p_{i,n,k}). \tag{4}$$

By leveraging heterogeneity as a heuristic for temperature initialization and implementing adaptive decay mechanisms, we can navigate the complex temperature optimization landscape more effectively than existing uniform approaches.

## 3 METHODOLOGY

### 3.1 OVERALL FRAMEWORK

Our proposed FedChill extends the FedAvg (McMahan et al., 2017) algorithm by introducing client-specific temperature scaling to handle statistical heterogeneity across clients. It consist of three key components i) Client-Specific Heterogeneity Score, ii) Personalized Temperature Initialization, and iii) Adaptive Temperature Decay Strategy. The overall algorithm is summarized in Appendix 1.

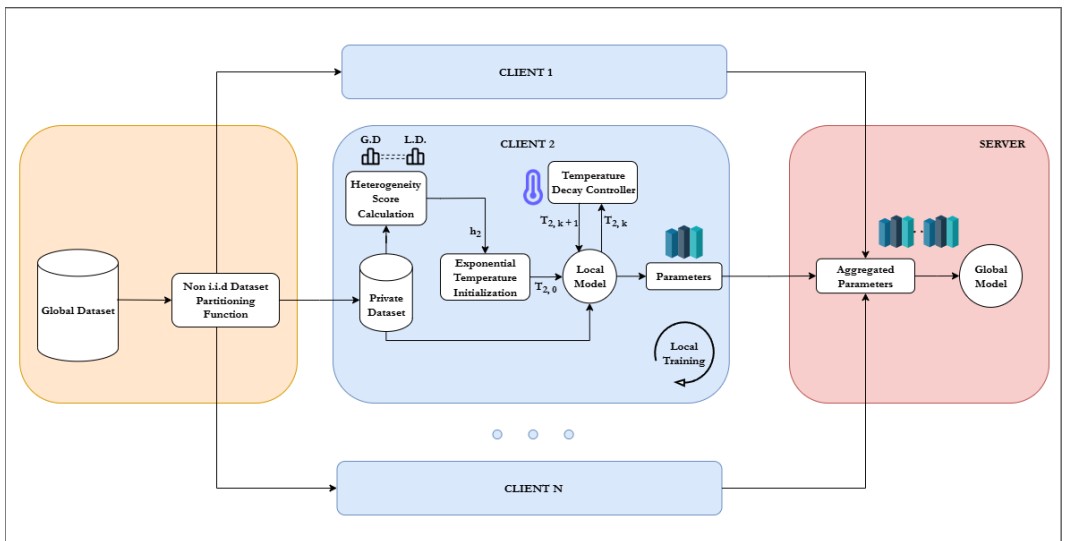

Figure 2: Overview of the proposed FedChill framework. (Key: $T_{n,k}$ : Temperature of client $n$ at communication round $k$, $h_n$ : Heterogeneity score of client $n$, G.D: Global Distribution, L.D.: Local Distribution).

Our framework in Fig. 2 begins by partitioning data among $n$ clients. Each client's local class distribution is compared against an approximated global distribution to compute a heterogeneity score. These heterogeneity scores are used to initialize personalized softmax temperatures through an exponential decay function before the communication rounds begin.

With client-specific temperatures are initialized, the global model (e.g. a lightweight convolutional neural network) is broadcast to all participating clients. After loading the latest global model, the clients begin training locally on their private datasets $\mathcal{D}_n$, wherein each client minimizes the negative log-likelihood loss:

$$\mathcal{L}_{n,k} = -\sum_{\forall i} \log\left(p_{i,n,k}\right). \tag{5}$$

Following training, each client evaluates its model on a private validation set. If performance stagnates, the local controller adaptively reduces the temperature by multiplying it with a fixed decay factor once a predefined tolerance threshold is exceeded, thereby intensifying the sharpness of predictions in subsequent rounds.

Finally, after all clients complete local training, model updates are aggregated on the server using FedAvg (McMahan et al., 2017). This process is then repeated for a fixed number of communication rounds.

### 3.2 CLIENT-SPECIFIC HETEROGENEITY SCORE

The temperature initialization process is driven by a client-specific heterogeneity score that quantifies how much a client's local data distribution diverges from the global distribution. To estimate the

global class distribution without sharing client data distributions, we use a principled approximation approach. The server constructs an ideal global distribution that represents a balanced allocation of classes across the federation. In a balanced classification scenario, this corresponds to a uniform distribution where each class has equal representation. For datasets with known natural class imbalances, the global distribution can be adjusted to reflect these expected priors. This approach preserves privacy while providing a meaningful reference point against which client heterogeneity can be measured.

For each client, we calculate the heterogeneity score by comparing its class distribution with this global distribution:

$$h_n = \sum_{c=1}^{C} \left| \frac{p_n(c)}{p_g(c)} - 1 \right| \tag{6}$$

where $p_i(c)$ is the probability of class $c$ in client $n$'s dataset, $p_g(c)$ is the probability of class $c$ in the global distribution, and $C$ is the number of classes. This score is normalized to the range $[0, 1]$, with higher values indicating greater divergence from the global distribution.

### 3.3 Personalized Temperature Initialization

The temperature is then initialized using an exponential decay function:

$$T_{n,0} = T_{\max} \cdot e^{-s \cdot h_n} \tag{7}$$

where $T_{\max}$ is the maximum allowed temperature, $s$ is a scaling factor, and $h_n$ is the client's heterogeneity score.

The use of an exponential decay policy ensures a smooth, non-linear scaling of temperature with respect to heterogeneity. This choice amplifies the impact of higher heterogeneity scores, allowing clients with significantly skewed data to receive much lower temperatures, thereby enforcing stronger learning signals. Conversely, clients with near-i.i.d. data maintain higher temperatures, preserving the expressiveness of their predictions.

### 3.4 Adaptive Temperature Decay Strategy

Our approach FedChill integrates an adaptive temperature decay mechanism (Appendix A.1, Algorithm 1) into the standard federated learning workflow to improve convergence under non-i.i.d. settings. During local training in each communication round, clients evaluate their models on private validation data at the end to compute a validation accuracy score. The client tracks its performance over time using past accuracies. If accuracy does not improve for $P$ consecutive rounds (i.e., stagnation occurs), the temperature is adaptively decayed as follows:

$$T_{n,k+1} \leftarrow \max(\gamma \cdot T_{n,k}, T_{\min}) \tag{8}$$

where $\gamma \in (0, 1)$ is a fixed decay factor and $T_{\min}$ is a lower bound to prevent excessive sharpening.

The condition for stagnation is checked by comparing the current validation accuracy with the accuracy two rounds ago. If the current accuracy is less than or equal to that value, a counter is incremented. Otherwise, the counter is reset. Once this counter reaches the patience threshold $P$, and the temperature is still sufficiently above the minimum threshold ($T_{n,k} > 1.1 \cdot T_{\min}$), the temperature is updated and the counter is reset.

This mechanism allows each client to autonomously calibrate its prediction confidence based on its performance trajectory, thereby adapting to local data distributions.

## 4 Experimental Setup

### 4.1 Dataset Selection

We utilize CIFAR-10, CIFAR-100 (Krizhevsky & Hinton, 2009), and SVHN (Netzer et al., 2011) datasets to evaluate our FedChill approach. These datasets were selected because they enable evaluation across different complexity levels (10 vs. 100 classes), allow meaningful simulation of non-

i.i.d distributions via Dirichlet partitioning. They also provide sufficient challenge while remaining computationally feasible, and serve as standard benchmarks in federated learning (Krizhevsky & Hinton, 2009; Li et al., 2024).

## 4.2 IMPLEMENTATION

### 4.2.1 MODEL ARCHITECTURE

We employ custom convolutional neural networks (CNNs) across our experiments, with approximately 685k, 4.96M, and 13.96M parameters respectively. For SVHN, we utilize an architecture of 1.243M. The architectures are outlined in Appendix A.2, Table 8. The architecture for SVHN is presented in Appendix A.2, Table 9.

### 4.2.2 DATA PARTITIONING

We implement a non-i.i.d partitioning strategy using Dirichlet allocation (Chen et al., 2023):

- For each class $c_i \in C$, we sample class proportions from $\text{Dir}(\alpha \cdot \mathbf{1})$

- The sampled proportions are normalized and adjusted to ensure balanced client datasets

- Each client receives a local training set and a validation set, the latter derived from the global test set

- A client-specific subsampling rate (`frac`) controls the local dataset size

We adopt a Dirichlet-based partitioning scheme to generate realistic heterogeneous data distributions among clients, with concentration parameters serving as the primary mechanism for controlling statistical heterogeneity levels (Chen et al., 2023).

### 4.2.3 EXPERIMENTAL SETTINGS

In our primary experimental evaluation, we implement FedChill using `PyTorch` (Paszke et al., 2019) with consistent configuration across all experiments. We employ the `SGD` optimizer (Ruder, 2017) with a learning rate of 0.01 and train for 5 local epochs per communication round, with a total of 50 communication rounds. For data partitioning, we utilize a Dirichlet distribution ($\alpha = 0.5$) to simulate non-i.i.d scenarios across varying client configurations (10, 20, and 50 clients) (Chen et al., 2023). The batch size is set to 64, and we use a data fraction of 0.1 for 10-client, 0.2 for 20-client, and 0.5 for 50-client configurations (Chen et al., 2023).

For FedChill's adaptive temperature mechanism, we initialize temperatures based on client heterogeneity scores, constrained within the range $[0.05, 1.0]$ (Lee et al., 2024). The scaling factor $s$ was chosen on the basis of an empirical experiment using a 1D hyperparameter sweep provided in Appendix A.4.1, and temperature decay factor $\gamma \in \{0.8, 0.95\}$ were selected based on empirical analysis across different experimental configurations. Temperature adjustments are triggered when performance plateaus for two consecutive rounds as shown in Appendix A.1.

## 4.3 EVALUATION METRICS

To comprehensively assess performance, we track three key aspects of the federated system. First, we measure the test accuracy of the global model on the global test set after each communication round (Chen et al., 2023). Second, we evaluate client performance by recording local validation accuracy on private data as well as generalization to the global test set (Chen et al., 2023). Finally, we monitor the evolution of temperature parameters across rounds to capture the adaptive behavior of our approach.

For comprehensive assessment, in our primary experiment we conduct experiments with varying numbers of clients (10, 20, and 50) and data fractions (0.1, 0.2, and 0.5) to evaluate the scalability and robustness of our approach under different federated learning scenarios (Chen et al., 2023).

## 5 RESULTS AND EVALUATION

### 5.1 COMPARISON WITH SOTA

Our experimental evaluation demonstrates FedChill's superior performance across 3 different dataset complexities, client configurations, and heterogeneity scenarios. Tables 1 and 2 present a comprehensive comparison with state-of-the-art federated learning methods.

Table 1: Comparison of local and global accuracy across multiple SOTA methods on CIFAR10, CIFAR100, and SVHN with varying clients

| Dataset | Scheme | Local Accuracy | | | Global Accuracy | | | Params (M) | Pub Data |
|---|---|---|---|---|---|---|---|---|---|
| | | 10 | 20 | 50 | 10 | 20 | 50 | | |
| CIFAR10 | FedAvg | 0.5950 | 0.6261 | 0.5825 | 0.4741 | 0.5516 | 0.3773 | 11.209 | No |
| | FedProx | 0.5981 | 0.6295 | 0.6490 | 0.4793 | 0.5258 | 0.5348 | 22.418 | No |
| | Moon | 0.5901 | 0.6482 | 0.5513 | 0.4579 | 0.5651 | 0.3514 | 33.627 | No |
| | FedAlign | 0.5946 | 0.6023 | 0.6402 | 0.4976 | 0.5184 | 0.5641 | 11.209 | No |
| | FedGen | 0.5879 | 0.6395 | 0.6533 | 0.4800 | 0.5408 | 0.5651 | 11.281 | No |
| | FedMD | 0.6147 | 0.6666 | 0.6533 | 0.5088 | 0.5575 | 0.5714 | 11.209 | Yes |
| | FedProto | 0.6131 | 0.6505 | 0.5939 | 0.5012 | 0.5548 | 0.4016 | 11.209 | No |
| | FedHKD* | 0.6227 | 0.6515 | 0.6675 | 0.5049 | 0.5596 | 0.5074 | 11.209 | No |
| | FedHKD | 0.6254 | 0.6816 | 0.6671 | 0.5213 | 0.5735 | 0.5493 | 11.209 | No |
| | **FedChill** | **0.6887** | **0.7239** | **0.7626** | **0.5335** | **0.6410** | **0.6820** | 4.959 | No |
| CIFAR100 | FedAvg | 0.2361 | 0.2625 | 0.2658 | 0.2131 | 0.2748 | 0.2907 | 11.215 | No |
| | FedProx | 0.2332 | 0.2814 | 0.2955 | 0.2267 | 0.2708 | 0.2898 | 22.430 | No |
| | Moon | 0.2353 | 0.2729 | 0.2428 | 0.2141 | 0.2652 | 0.1928 | 33.645 | No |
| | FedAlign | 0.2467 | 0.2617 | 0.2854 | 0.2281 | 0.2729 | 0.2933 | 11.215 | No |
| | FedGen | 0.2393 | 0.2701 | 0.2739 | 0.2176 | 0.2620 | 0.2739 | 11.333 | No |
| | FedMD | 0.2681 | 0.3054 | 0.3293 | 0.2323 | 0.2669 | 0.2968 | 11.215 | Yes |
| | FedProto | 0.2568 | 0.3188 | 0.3170 | 0.2121 | 0.2756 | 0.2805 | 11.215 | No |
| | FedHKD* | 0.2551 | 0.2997 | 0.3016 | 0.2286 | 0.2715 | 0.2976 | 11.215 | No |
| | FedHKD | **0.2981** | **0.3245** | 0.3375 | 0.2369 | 0.2795 | 0.2988 | 11.215 | No |
| | **FedChill** | 0.2412 | 0.3025 | **0.4185** | **0.2619** | **0.3266** | **0.3823** | 4.959 | No |
| SVHN | FedAvg | 0.6766 | 0.7329 | 0.6544 | 0.4948 | 0.6364 | 0.5658 | 1.286 | No |
| | FedProx | 0.6927 | 0.6717 | 0.6991 | 0.5191 | 0.6419 | 0.6139 | 2.572 | No |
| | Moon | 0.6602 | 0.7085 | 0.7192 | 0.4883 | 0.5536 | 0.6543 | 3.858 | No |
| | FedAlign | 0.7675 | 0.7920 | 0.7656 | 0.6426 | 0.7138 | 0.7437 | 1.286 | No |
| | FedGen | 0.5788 | 0.5658 | 0.4679 | 0.3622 | 0.3421 | 0.3034 | 1.357 | No |
| | FedMD | 0.8038 | 0.8086 | 0.7912 | 0.6812 | 0.7344 | 0.8085 | 1.286 | Yes |
| | FedProto | 0.8071 | 0.8148 | 0.8039 | 0.6064 | 0.6259 | 0.7895 | 1.286 | No |
| | FedHKD* | 0.8064 | 0.8157 | 0.8072 | 0.6405 | 0.6884 | 0.7921 | 1.286 | No |
| | FedHKD | 0.8086 | 0.8381 | 0.7891 | 0.6781 | 0.7357 | 0.7891 | 1.286 | No |
| | **FedChill** | **0.8654** | **0.8884** | **0.9022** | **0.8649** | **0.8723** | **0.9125** | 1.243 | No |

### 5.1.1 BASELINES

We compare FedChill with several state-of-the-art federated learning methods including FedAvg (McMahan et al., 2017), FedProx (Li et al., 2020), Moon (Li et al., 2021), FedAlign (Mendieta et al., 2022), FedGen (Zhu et al., 2021), FedMD (Li & Wang, 2019), and FedHKD (Chen et al., 2023). The novelty of FedChill lies in its adaptive temperature-based regularization mechanism that requires no public dataset, generative model, or additional communication overhead. This contrasts with methods like FedMD which relies on a public dataset, and FedGen which employs a generative model. While FedHKD achieves strong performance through knowledge distillation, our approach differs by using client-specific temperature scaling based on our heterogeneity metrics.

### 5.1.2 PERFORMANCE ANALYSIS

Our experimental evaluation demonstrates FedChill's superior performance across different dataset complexities, client configurations, and heterogeneity scenarios. Table 1 presents a comprehensive comparison with state-of-the-art federated learning methods.

**Global Model Performance:** On CIFAR-10, FedChill achieves significant improvements in global model accuracy across all client configurations. With 10 clients, FedChill attains 53.35% accuracy, outperforming FedHKD (52.13%) and FedMD (50.88%). This advantage becomes more pronounced with 20 clients (64.10% vs. 57.35%) and 50 clients (68.20% vs. 54.93%), demonstrating exceptional scalability. Similarly, on the more challenging CIFAR-100, FedChill delivers superior global accuracy, particularly with larger client numbers (38.23% vs. 29.88% with 50 clients), highlighting its effectiveness on complex tasks. On SVHN, FedChill achieves 86.49%, 87.23%, and 91.25% global accuracy for 10, 20, and 50 clients, respectively, highlighting its robustness on real-world image datasets.

**Local Model Performance:** FedChill exhibits remarkable improvement in client-side model performance. On CIFAR-10, it achieves local accuracies of 68.87%, 72.39%, and 76.26% for 10, 20, and 50 clients, respectively, substantially outperforming all baselines. This pattern extends to CIFAR-100 for configurations with larger client numbers, where FedChill reaches 41.85% local accuracy with 50 clients, compared to FedHKD's 33.75%. On SVHN, FedChill achieves 90.22% local accuracy for 50 clients, above the best baseline. The positive correlation between client count and performance improvement is particularly noteworthy, suggesting that FedChill effectively leverages client diversity.

**Model Efficiency:** A key advantage of FedChill is its parameter efficiency, utilizing only 4.96M parameters compared to 11.21M for most baselines (and 22.43M for FedProx, 33.65M for Moon) (Chen et al., 2023). This represents a 55.7% reduction in model size while achieving superior performance, leading to reduced communication overhead and computational requirements in resource-constrained federated environments.

**Heterogeneity Handling:** Table 2 provides further evidence of FedChill's effectiveness in handling data heterogeneity. Under both low ($\alpha = 0.2$) and high ($\alpha = 5$) Dirichlet concentration parameters, FedChill consistently outperforms all baselines. The margin is particularly significant in the challenging low-concentration setting ($\alpha = 0.2$), where FedChill achieves 80.77% local and 50.85% global accuracy, compared to the next best method's 67.89% and 47.36%, respectively. This demonstrates FedChill's robust adaptation to varying levels of statistical heterogeneity.

Table 2: CIFAR-10 local and global accuracy under varying heterogeneity ($\alpha$)

| Method | Local Acc. | | Global Acc. | |
|---|---|---|---|---|
| | $\alpha$=0.2 | $\alpha$=5 | $\alpha$=0.2 | $\alpha$=5 |
| FedAvg | 0.5917 | 0.4679 | 0.3251 | 0.5483 |
| FedProx | 0.6268 | 0.4731 | 0.3845 | 0.5521 |
| Moon | 0.5762 | 0.3794 | 0.3229 | 0.4256 |
| FedAlign | 0.6434 | 0.4799 | 0.4446 | 0.5526 |
| FedGen | 0.6212 | 0.4432 | 0.4623 | 0.4432 |
| FedMD | 0.6532 | 0.4940 | 0.4408 | 0.5543 |
| FedProto | 0.6471 | 0.4802 | 0.3887 | 0.5488 |
| FedHKD | 0.6789 | 0.4976 | 0.4736 | 0.5573 |
| **FedChill** | **0.8077** | **0.5995** | **0.5085** | **0.6435** |

## 5.2 ABLATION STUDIES

We conduct comprehensive ablation studies to evaluate: (1) the impact of temperature scaling across different model capacities, and (2) the individual contributions of FedChill's key components across varying levels of data heterogeneity.

### 5.2.1 TEMPERATURE SCALING ACROSS MODEL CAPACITIES

Table 3 presents accuracy results for three CNN architectures (see Appendix A.2) with different parameter counts (685K, 4.95M, and 13.95M) across various temperature settings and our adaptive FedChill approach. Temperature values significantly affect performance across all architectures, with $T = 0.25$ generally outperforming $T = 1.0$ (standard cross-entropy) by 2–4% in both local and global accuracy, confirming our hypothesis that appropriate scaling benefits training in federated settings. While fixed temperatures vary in effectiveness across architectures, FedChill consistently achieves superior performance, yielding the highest local accuracy for CNN1 (67.16% vs. 66.46%) and CNN2 (67.56% vs. 67.87%), while remaining competitive for CNN3. Moreover, model capacity strongly influences temperature sensitivity: the largest model (CNN3) exhibits the greatest variance in performance (6.73% difference between best and worst local accuracy), indicating that higher-capacity models are more prone to temperature effects, likely due to their greater tendency to overfit to local data distributions.

Table 3: Ablation Study: Local vs global accuracy for different CNNs and frameworks on CIFAR-10

| Architecture | Framework | Local Acc (%) | Global Acc (%) |
|---|---|---|---|
| CNN 1 (685k) | Flex&Chill T=0.05 | 63.83 | 47.36 |
| | Flex&Chill T=0.25 | 66.46 | 48.66 |
| | Flex&Chill T=0.5 | 66.46 | 47.76 |
| | Flex&Chill T=1.0 | 63.44 | 46.76 |
| | FedChill | 67.16 | 50.14 |
| CNN 2 (4.95M) | Flex&Chill T=0.05 | 64.91 | 48.94 |
| | Flex&Chill T=0.25 | 67.87 | 50.08 |
| | Flex&Chill T=0.5 | 66.13 | 49.44 |
| | Flex&Chill T=1.0 | 64.28 | 48.82 |
| | FedChill | 67.56 | 52.03 |
| CNN 3 (13.95M) | Flex&Chill T=0.05 | 63.44 | 42.43 |
| | Flex&Chill T=0.25 | 69.17 | 49.00 |
| | Flex&Chill T=0.5 | 68.14 | 48.12 |
| | Flex&Chill T=1.0 | 65.42 | 45.75 |
| | FedChill | 67.04 | 47.58 |

Table 4: Ablation Study: Showing component contributions in FedChill (30 training rounds, 10 clients, $frac = 0.1$)

| Het. ($\alpha$) | CNN Arch. | Component | Local Acc (%) | Global Acc (%) |
|---|---|---|---|---|
| $\alpha = 0.5$ | CNN1 (0.685M) | FedAvg (T=1.0) | 64.54 | 46.96 |
| | | Fixed T=0.05 | 65.65 | 47.26 |
| | | FedChill | 67.91 | 49.33 |
| | | FedChill* | 65.36 | 49.49 |
| | CNN3 (13.95M) | FedAvg (T=1.0) | 65.97 | 45.74 |
| | | Fixed T=0.05 | 61.73 | 39.22 |
| | | FedChill | 67.69 | 47.55 |
| | | FedChill* | 64.55 | 45.39 |
| $\alpha = 5.0$ | CNN1 (0.685M) | FedAvg (T=1.0) | 52.08 | 54.56 |
| | | Fixed T=0.05 | 50.55 | 56.10 |
| | | FedChill | 54.27 | 55.67 |
| | | FedChill* | 53.39 | 55.81 |
| | CNN3 (13.95M) | FedAvg (T=1.0) | 50.70 | 53.44 |
| | | Fixed T=0.05 | 47.24 | 53.48 |
| | | FedChill | 53.85 | 55.57 |
| | | FedChill* | 50.81 | 56.54 |

### 5.2.2 COMPONENT-WISE CONTRIBUTION ANALYSIS

Table 4 isolates the contributions of heterogeneity-based initialization versus adaptive decay across high ($\alpha = 0.5$) and low ($\alpha = 5.0$) heterogeneity. In high heterogeneity settings, initialization serves as the primary driver, boosting local accuracy by up to 3.37% over baselines and effectively adapting to skewed data. Conversely, in near-homogeneous settings ($\alpha = 5.0$), the full adaptive decay (FedChill*) yields superior global accuracy, suggesting that dynamic decay prevents overfitting when client distributions are uniform. Crucially, both adaptive variants consistently outperform fixed static temperatures ($T = 0.05$), validating that client-specific scaling is superior to uniform regularization strategies.

**Gradient Analysis: Magnitude vs. Direction**   To address whether FedChill functions merely as an adaptive learning rate, we dissected the mechanism by isolating the effects of gradient magnitude and direction. We compared the full FedChill method against a **Magnitude-Only** variant (standard softmax, manually scaled gradients) and a **Direction-Only** variant (temperature-scaled softmax, normalized magnitudes).

| Method | Final Accuracy | $\Delta$ vs Baseline |
|---|---|---|
| FedAvg-Baseline | 55.19% | — |
| FedChill-MagOnly | 59.68% | **+4.49%** |
| FedChill-DirOnly | 52.09% | -3.10% |
| FedChill-Full | 57.78% | +2.59% |

Table 5: Decoupling gradient magnitude and direction effects.

Results in Table 5 reveal that FedChill acts primarily through **gradient magnitude amplification**. Interestingly, directional changes induced by low temperatures actually degrade performance when isolated (-3.10%). However, the full FedChill configuration yields positive gains, suggesting a constructive interaction where magnitude amplification allows heterogeneous clients to retain influence during aggregation, while temperature-controlled directional adjustments prevent excessive divergence.

### 5.2.3 ROBUSTNESS TO FEATURE SHIFT

To assess FedChill under feature shift (concept drift), we conducted experiments on CIFAR-10 where data is distributed IID but subjected to client-specific Gaussian noise ($\sigma \in [0.1, 0.3]$). As shown in Table 6, FedChill achieves 58.92% accuracy compared to FedAvg's 57.98%, demonstrating robustness even when heterogeneity stems from feature skew rather than label skew.

| Method | Accuracy (%) |
|---|---|
| FedAvg | 57.98 |
| **FedChill** | **58.92** |

Table 6: Performance under feature shift (IID + Gaussian noise).

### 5.2.4 HYPERPARAMETER SENSITIVITY ($P$ AND $s$)

We analyzed the sensitivity of the patience parameter $P$ and scaling factor $s$. As shown in Table 7, performance is more sensitive to $s$ than to $P$, as $s$ directly determines the initial temperature scale and gradient magnitudes. Values of $s \approx 2$ consistently produce the highest accuracy across varying patience levels. While lower $P$ values provide moderate improvements, decreasing $P$ yields limited marginal gains once $s$ is within the optimal range of $[2, 2.5]$.

To validate that this parameter choice is not specific to a single configuration, we provide an extended analysis in Appendix A.4.1 (Table 10). This broader sweep confirms that $s = 2.0$ yields robust performance across varying client counts (10, 20, and 50) and also lists performance against other heterogeneity metrics like KL-divergence in extreme-scaling regimes (Appendix A.4.2).

| **Patience** ($P$) | $s = 1.0$ | $s = 2.0$ | $s = 3.0$ | $s = 4.0$ | $s = 5.0$ |
|---|---|---|---|---|---|
| 1 | 54.58 | **57.11** | 52.96 | 51.50 | 53.06 |
| 3 | 54.86 | **56.80** | 50.91 | 49.86 | 52.79 |
| 5 | 55.37 | **55.92** | 49.10 | 50.92 | 50.49 |
| 10 | 55.25 | **56.68** | 50.83 | 49.91 | 51.29 |

Table 7: Sensitivity analysis of scaling factor ($s$) and patience ($P$) on server accuracy (%).

## 6 CONCLUSION

In this work, we presented **FedChill**, a heterogeneity-aware framework that addresses the critical challenge of statistical divergence in federated learning. While prior approaches have utilized static temperature scaling to sharpen local objectives (Lee et al., 2024), we demonstrate that a one-size-fits-all strategy is suboptimal in non-i.i.d. environments. FedChill overcomes this by introducing a dynamic, context-aware strategy rooted in two key mechanisms: a privacy-preserving heterogeneity score for personalized temperature initialization, and a performance-aware decay schedule that adapts to training stagnation.

Our extensive evaluation against 8 state-of-the-art methods confirms the efficacy of this two-fold strategy. On the challenging CIFAR-100 dataset with 50 clients, FedChill yielded improvements of $8.1\%$ in local accuracy and $8.35\%$ in global accuracy compared to baselines. Notably, these gains are achieved with a highly efficient architecture, requiring $2.26\times$ fewer parameters than comparable state-of-the-art methods.

Finally, we highlight the extensibility of our approach. Since FedChill operates exclusively by modulating the local training objective, it is independent to server-side aggregation logic. Future work can explore synergistic integrations with aggregation-level strategies, such as FedProx or FedAlign, to simultaneously address heterogeneity at both the local optimization and global model-fusion stages.

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

## A  APPENDIX

### A.1  ADAPTIVE TEMPERATURE DECAY ALGORITHM

---

**Algorithm 1** Adaptive Temperature Decay in Client Training

---

**Require:**
    Validation Accuracy History $\mathcal{A}_n$,
    Current Temperature $T_{n,k}$,
    Patience Threshold $P$,
    Decay Factor $\gamma$,
    Minimum Temperature $T_{\min}$

 1: *stagnation_count* $\leftarrow 0$
 2: $\mathcal{A}_n \leftarrow [\,]$
 3: **for each communication round** $k$ **do**
 4:    $a_k \leftarrow$ current round's validation accuracy
 5:    Append $a_k$ to $\mathcal{A}_n$
 6:
 7:    **if** $|\mathcal{A}_n| \geq 3$ **and** $a_k \leq \mathcal{A}_n[-2]$ **then**
 8:       *stagnation_count* $\leftarrow$ *stagnation_count* $+ 1$
 9:    **else**
10:       *stagnation_count* $\leftarrow 0$
11:    **end if**
12:
13:    **if** *stagnation_count* $\geq P$ **and** $T_{n,k} > 1.1 \cdot T_{\min}$ **then**
14:       $T_{n,k+1} \leftarrow \max(\gamma \cdot T_{n,k}, T_{\min})$
15:       *stagnation_count* $\leftarrow 0$
16:    **else**
17:       $T_{n,k+1} \leftarrow T_{n,k}$
18:    **end if**
19:
20: **end for**

---

## A.2 CNN ARCHITECTURES

Table 8: CNN architecture details (Key: Dims.: Dimensions, Padd.: Padding, Act.: Activation, C: 10 for CIFAR-10, SVHN & 100 for CIFAR-100)

| | CNN1 | | | | | | CNN2 | | | | | | CNN3 | | | | |
|---|---|---|---|---|---|---|---|---|---|---|---|---|---|---|---|---|---|
| # | # Type | Output Dims. | P. | Norm. | Act. | Others | # Type | Output Dims. | P. | Norm. | Act. | Others | # Type | Output Dims. | P. | Norm. | Act. | Others |
| 1 | 1 Conv2d $3 \times 32 \times 3$ | | 1 | BN | ReLU | - | 1 Conv2d $3 \times 64 \times 3$ 1 BN ReLU MaxPool(2,2) | | | | | | 1 Conv2d $3 \times 80 \times 3$ 1 BN ReLU - | | | | |
| 2 | 2 Conv2d $32 \times 32 \times 3$ | | 1 | BN | ReLU | MaxPool(2,2) | 2 Conv2d $64 \times 128 \times 3$ 1 BN ReLU MaxPool(2,2) | | | | | | 2 Conv2d $80 \times 80 \times 3$ 1 BN ReLU - | | | | |
| 3 | 3 Conv2d $32 \times 64 \times 3$ | | 1 | BN | ReLU | - | 3 Conv2d $128 \times 256 \times 3$ 1 BN ReLU MaxPool(2,2) | | | | | | 3 Conv2d $80 \times 80 \times 3$ 1 BN ReLU MaxPool(2,2) | | | | |
| 4 | 4 Conv2d $64 \times 64 \times 3$ | | 1 | BN | ReLU | MaxPool(2,2) | 4 Conv2d $256 \times 512 \times 3$ 1 BN ReLU Ada.AvgPool(1,1) | | | | | | 4 Conv2d $80 \times 160 \times 3$ 1 BN ReLU - | | | | |
| 5 | 5 Conv2d $64 \times 128 \times 3$ | | 1 | BN | ReLU | - | 5 FC 1024 - - ReLU Dropout(0.25) | | | | | | 5 Conv2d $160 \times 160 \times 3$ 1 BN ReLU - | | | | |
| 6 | 6 Conv2d $128 \times 128 \times 3$ | | 1 | BN | ReLU | Ada.AvgPool(2,2) | 6 FC 512 - - ReLU Dropout(0.25) | | | | | | 6 Conv2d $160 \times 160 \times 3$ 1 BN ReLU MaxPool(2,2) | | | | |
| 7 | 7 FC 512 | | - | BN | ReLU | Dropout(0.5) | 7 Output $C$ - - - - | | | | | | 7 Conv2d $160 \times 320 \times 3$ 1 BN ReLU - | | | | |
| 8 | 8 FC 256 | | - | BN | ReLU | Dropout(0.3) | | | | | | | 8 Conv2d $320 \times 320 \times 3$ 1 BN ReLU - | | | | |
| 9 | 9 Output $C$ | | - | - | - | - | | | | | | | 9 Conv2d $320 \times 320 \times 3$ 1 BN ReLU MaxPool(2,2) | | | | |
| | | | | | | | | | | | | | 10 Conv2d $320 \times 640 \times 3$ 1 BN ReLU - | | | | |
| | | | | | | | | | | | | | 11 Conv2d $640 \times 640 \times 3$ 1 BN ReLU Ada.AvgPool(2,2) | | | | |
| | | | | | | | | | | | | | 12 FC 1536 - BN ReLU Dropout(0.5) | | | | |
| | | | | | | | | | | | | | 13 FC 768 - BN ReLU Dropout(0.4) | | | | |
| | | | | | | | | | | | | | 14 FC 384 - BN ReLU Dropout(0.3) | | | | |
| | | | | | | | | | | | | | 15 Output $C$ - - - - | | | | |

Table 9: SVHN CNN architecture (Key: Dims.: Dimensions, Padd.: Padding, Act.: Activation, C: 10 for SVHN)

| # | Layer | Output Dim. | P. | Norm. | Act. | Others |
|---|---|---|---|---|---|---|
| 1 | Conv1 | $32 \times 32 \times 32$ | 1 | BN | ReLU | - |
| 2 | Conv2 | $32 \times 32 \times 32$ | 1 | BN | ReLU | MaxPool(2,2)$\rightarrow 32 \times 16 \times 16$ |
| 3 | Conv3 | $64 \times 16 \times 16$ | 1 | BN | ReLU | - |
| 4 | Conv4 | $64 \times 16 \times 16$ | 1 | BN | ReLU | MaxPool(2,2)$\rightarrow 64 \times 8 \times 8$ |
| 5 | Conv5 | $128 \times 8 \times 8$ | 1 | BN | ReLU | - |
| 6 | Conv6 | $128 \times 8 \times 8$ | 1 | BN | ReLU | MaxPool(2,2)$\rightarrow 128 \times 4 \times 4$ |
| 7 | Conv7 | $256 \times 4 \times 4$ | 1 | BN | ReLU | AdaptiveAvgPool(2,2)$\rightarrow 256 \times 2 \times 2$ |
| 8 | FC1 | 512 | - | BN | ReLU | Dropout(0.5) |
| 9 | FC2 | 256 | - | BN | ReLU | Dropout(0.3) |
| 10 | FC3 | $C$ | - | - | - | Logits |
| 11 | Output | $C$ | - | - | Softmax | Final prediction |

## A.3 CONVERGENCE ANALYSIS - FULL DEVICE PARTICIPATION

### ASSUMPTIONS

We analyze convergence under the following assumptions:

(A1) Temperature bounds: $0 < T_{\min} \leq T_k(t) \leq T_{\max} < \infty$.

(A2) $F_t$ is $L$-smooth for all $t$, with $L = O(1/T_{\min}^2)$.

(A3) Stochastic gradients have bounded variance: $\mathbb{E}\|\nabla f_i(w; \xi) - \nabla f_i(w)\|^2 \leq \sigma^2$.

(A4) Bounded heterogeneity: $\frac{1}{K} \sum_k \|\nabla F_k(w) - \nabla F(w)\|^2 \leq \zeta^2$.

(A5) Gradients bounded: $\|\nabla F_k(w)\| \leq G$.

(A6) Each client's temperature schedule changes finitely many times, say $M < \infty$.

**Proof for (A6).** Given that there are $T$ communication rounds and the patience factor for the FedChill algorithm is $n$ (i.e., the number of rounds without improvement after which the temperature is decayed), the maximum number of changes can only be $\lfloor T/n \rfloor$. Since $T$ is finite, the number of changes is also finite, which proves the claim.

### NOTATION

- $N$ clients, weights $p_k \geq 0$ with $\sum_{k=1}^{K} p_k = 1$.
- At round $t$ the server holds $w_t \in \mathbb{R}^d$.

- Each client $k$ has a population loss with temperature $T$:
$$F_k^T(w) = \mathbb{E}_{\mathcal{D}_k}[f_k^T(w)]$$
  where $f_k^T$ is the per-sample cross-entropy loss with temperature $T$.

- The **round-$t$ global objective** is:
$$F_t(w) := \sum_{k=1}^{K} p_k F_k^{T_{t,k}}(w).$$
  where $T_{t,k}$ is the temperature for client $k$ at round $t$.

FEDCHILL ALGORITHM

For $t = 0, 1, 2, \ldots$:

1. Server sends $w_t$ to clients.
2. Each client $k$ sets $w_{t,0}^k = w_t$ and runs $\tau$ steps of stochastic gradient descent with stepsize $\eta$:
$$w_{t,j+1}^k = w_{t,j}^k - \eta g_{t,j}^k, \qquad j = 0, \ldots, \tau - 1,$$
   where $g_{t,j}^k$ is an unbiased stochastic gradient of $F_k^{T_{t,k}}$.
3. Client returns $w_{t,\tau}^k$; server aggregates
$$w_{t+1} = \sum_{k=1}^{K} p_k w_{t,\tau}^k.$$

4. Each client may update its temperature $T_{t,k} \to T_{t+1,k}$.

PROOF

Since $F_t$ is $L$-smooth, for any $w, u$ we have
$$F_t(u) \le F_t(w) + \langle \nabla F_t(w), u - w \rangle + \frac{L}{2}\|u - w\|^2. \tag{9}$$

Let $w_t$ be the global model at round $t$. Each client $k$ performs $\tau$ steps of SGD with step size $\eta$:
$$w_{t,\tau}^k = w_t - \eta \sum_{s=0}^{\tau-1} g_{t,s}^k,$$

where $g_{t,s}^k$ is the stochastic gradient on client $k$ at local step $s$.

The server averages ($p_k = \frac{1}{K}$):
$$w_{t+1} = \frac{1}{K} \sum_{k=1}^{K} w_{t,\tau}^k. \tag{10}$$

Define the global update:
$$\Delta_t := w_{t+1} - w_t = -\eta \cdot \frac{1}{K} \sum_{k=1}^{K} \sum_{s=0}^{\tau-1} g_{t,s}^k.$$

Using (9) with $u = w_{t+1}$ and $w = w_t$ to be used for future reference:
$$F_t(w_{t+1}) \le F_t(w_t) + \langle \nabla F_t(w_t), \Delta_t \rangle + \frac{L}{2}\|\Delta_t\|^2. \tag{11}$$

Since each client uses the full dataset at every local step, the gradient is exact (no stochastic noise). Thus,
$$g_{t,s}^k = \nabla F_k(w_{t,s}^k).$$

Therefore, the global update becomes the following:

$$\Delta_t = -\eta \cdot \frac{1}{K} \sum_{k=1}^{K} \sum_{s=0}^{\tau-1} \nabla F_k(w_{t,s}^k). \tag{12}$$

Since each client uses its full dataset, there is no stochastic gradient noise. Thus, taking expectation only over client sampling (if partial participation) yields

$$\mathbb{E}\langle \nabla F_t(w_t), \Delta_t \rangle = -\eta\tau \|\nabla F_t(w_t)\|^2 + \underbrace{O(\eta\tau\zeta)}_{\text{client drift}}. \tag{13}$$

**Working:** Starting from the global update,
$$\Delta_t = -\eta \cdot \frac{1}{K} \sum_{k=1}^{K} \sum_{s=0}^{\tau-1} \nabla F_k(w_{t,s}^k),$$

we compare the average of local gradients to the global gradient. If all local iterates remained at $w_t$, we would have
$$\frac{1}{K} \sum_{k=1}^{K} \sum_{s=0}^{\tau-1} \nabla F_k(w_t) = \tau \cdot \nabla F_t(w_t).$$

Define the client drift error:
$$\delta_t := \frac{1}{K} \sum_{k=1}^{K} \sum_{s=0}^{\tau-1} \left[ \nabla F_k(w_{t,s}^k) - \nabla F_k(w_t) \right].$$

Thus, the decomposition becomes
$$\frac{1}{K} \sum_{k=1}^{K} \sum_{s=0}^{\tau-1} \nabla F_k(w_{t,s}^k) = \tau \cdot \nabla F_t(w_t) + \delta_t,$$

and therefore
$$\Delta_t = -\eta\big(\tau \nabla F_t(w_t) + \delta_t\big).$$

The descent term is
$$\langle \nabla F_t(w_t), \Delta_t \rangle = -\eta\tau \|\nabla F_t(w_t)\|^2 - \eta\langle \nabla F_t(w_t), \delta_t \rangle.$$

Using Cauchy–Schwarz and assuming $\|\delta_t\| \le \tau\zeta$, we obtain
$$\langle \nabla F_t(w_t), \Delta_t \rangle \le -\eta\tau \|\nabla F_t(w_t)\|^2 + \eta\tau\zeta \|\nabla F_t(w_t)\|.$$

Taking expectation gives the bound
$$\mathbb{E}\langle \nabla F_t(w_t), \Delta_t \rangle \le -\eta\tau \|\nabla F_t(w_t)\|^2 + O(\eta\tau\zeta).$$

Similarly, the quadratic term satisfies
$$\mathbb{E}\|\Delta_t\|^2 = O\big(\eta^2\tau^2(\zeta^2 + G^2)\big), \tag{14}$$

where $\zeta$ quantifies data heterogeneity across clients and $G$ bounds the gradient norm.

**Quadratic term bound.** Recall the decomposition
$$\Delta_t = -\eta\Big(\tau \nabla F_t(w_t) + \delta_t\Big), \qquad \delta_t := \frac{1}{K} \sum_{k=1}^{K} \sum_{s=0}^{\tau-1} \big(\nabla F_k(w_{t,s}^k) - \nabla F_k(w_t)\big).$$

We first expand the norm square:
$$\|\Delta_t\|^2 = \eta^2 \big\|\tau \nabla F_t(w_t) + \delta_t\big\|^2$$
$$\le 2\eta^2\Big(\tau^2 \|\nabla F_t(w_t)\|^2 + \|\delta_t\|^2\Big),$$

where we used $\|a + b\|^2 \le 2\|a\|^2 + 2\|b\|^2$.

Next we bound $\|\delta_t\|^2$. Using the inequality $\left\|\frac{1}{m}\sum_{i=1}^{m} v_i\right\|^2 \le \frac{1}{m}\sum_{i=1}^{m}\|v_i\|^2$ with $m = K\tau$ (there are $K\tau$ terms in the definition of $\delta_t$), we get

$$\|\delta_t\|^2 \le \frac{1}{K\tau}\sum_{k=1}^{K}\sum_{s=0}^{\tau-1}\left\|\nabla F_k(w_{t,s}^k) - \nabla F_k(w_t)\right\|^2.$$

Introduce the per-round heterogeneity measure

$$\zeta_t^2 := \frac{1}{K\tau}\sum_{k=1}^{K}\sum_{s=0}^{\tau-1}\left\|\nabla F_k(w_{t,s}^k) - \nabla F_t(w_t)\right\|^2,$$

and note the identity
$$\nabla F_k(w_{t,s}^k) - \nabla F_k(w_t) = \big(\nabla F_k(w_{t,s}^k) - \nabla F_t(w_t)\big) - \big(\nabla F_k(w_t) - \nabla F_t(w_t)\big).$$

By expanding squared norms and using the triangle / Jensen inequalities one obtains (up to constant factors) a bound of the form

$$\|\delta_t\|^2 \le C_1\tau\,\zeta_t^2 + C_2\tau^2 \cdot \frac{1}{K}\sum_{k=1}^{K}\|\nabla F_k(w_t) - \nabla F_t(w_t)\|^2,$$

and under the common bounded-dissimilarity assumption

$$\frac{1}{K}\sum_{k=1}^{K}\|\nabla F_k(w) - \nabla F(w)\|^2 \le \zeta^2 \quad \text{for all } w,$$

this simplifies (absorbing constants) to
$$\|\delta_t\|^2 \le C\,\tau\,\zeta^2,$$

for some universal constant $C$ (we may take $C = 1$ with the more careful definition of $\zeta_t$ used earlier). For a simple, clean bound it suffices to use
$$\|\delta_t\|^2 \le \tau\,\zeta_t^2 \le \tau\,\zeta^2.$$

Combining the two bounds gives
$$\|\Delta_t\|^2 \le 2\eta^2\Big(\tau^2\|\nabla F_t(w_t)\|^2 + \tau\,\zeta^2\Big).$$

Finally, taking expectation (over client sampling, if any) and using a uniform gradient bound $\|\nabla F_t(w_t)\| \le G$ if desired, we obtain the commonly stated form

$$\boxed{\mathbb{E}\|\Delta_t\|^2 = O\big(\eta^2\tau^2(\mathbb{E}\|\nabla F_t(w_t)\|^2 + \zeta^2)\big) = O\big(\eta^2\tau^2(\zeta^2 + G^2)\big).}$$

**Descent bound.** Taking expectation and applying the results from the previous results:

*(i) Descent term.*
$$\mathbb{E}\langle\nabla F_t(w_t), \Delta_t\rangle \le -\eta\tau\|\nabla F_t(w_t)\|^2 + O(\eta\tau\zeta).$$

*(ii) Quadratic term.* From the quadratic bound,
$$\tfrac{L}{2}\mathbb{E}\|\Delta_t\|^2 = O\big(L\eta^2\tau^2(\zeta^2 + G^2)\big).$$

Combining (i) and (ii) in equation 11 gives
$$\mathbb{E}[F_t(w_{t+1})] \le F_t(w_t) - \eta\tau\|\nabla F_t(w_t)\|^2 + O(\eta\tau\zeta) + O\big(L\eta^2\tau^2(\zeta^2 + G^2)\big). \tag{15}$$

**Absorbing error terms.** Since there is no variance term (full-batch), the error contribution is only due to heterogeneity and higher-order smoothness. For sufficiently small $\eta$,
$$O(\eta\tau\zeta) + O\big(L\eta^2\tau^2(\zeta^2 + G^2)\big) \le \tfrac{1}{4}\eta\tau\|\nabla F_t(w_t)\|^2 + O(\eta^3 L^2\tau^3).$$

**Final descent inequality.** Thus, under full-batch gradients,
$$F_t(w_{t+1}) \le F_t(w_t) - \tfrac{3}{4}\eta\tau\|\nabla F_t(w_t)\|^2 + O(\eta^3 L^2\tau^3). \tag{16}$$

Recall the per-round descent inequality under full-batch gradients (cf. equation 16):

$$F_t(w_{t+1}) \leq F_t(w_t) - \tfrac{3}{4}\eta\tau\|\nabla F_t(w_t)\|^2 + O(\eta^3 L^2 \tau^3).$$

When temperature changes from $T_{t,k}$ to $T_{t+1,k}$, the loss shifts from $F_t$ to $F_{t+1}$.

Define the objective shift

$$\Delta F_t(w) := F_{t+1}(w) - F_t(w),$$

and assume (A6) that the temperature (hence the objective) changes at most $M$ rounds and that along the trajectory

$$|\Delta F_t(w_t)| \leq \Delta_{\max} \quad \text{for all } t.$$

Therefore

$$\sum_{t=0}^{T-1} |\Delta F_t(w_t)| \leq M\Delta_{\max}.$$

Add the shift to convert the bound for $F_t(w_{t+1})$ into a bound for $F_{t+1}(w_{t+1})$:

$$F_{t+1}(w_{t+1}) = F_t(w_{t+1}) + \Delta F_t(w_{t+1}) \leq F_t(w_t) - \tfrac{3}{4}\eta\tau\|\nabla F_t(w_t)\|^2 + O(\eta^3 L^2 \tau^3) + \Delta F_t(w_{t+1}).$$

Sum the last inequality over $t = 0, \dots, T-1$. The left-hand side telescopes:

$$\sum_{t=0}^{T-1} \big(F_{t+1}(w_{t+1}) - F_t(w_t)\big) = F_T(w_T) - F_0(w_0).$$

Rearranging and summing the error / shift terms yields

$$\frac{3}{4}\eta\tau \sum_{t=0}^{T-1} \|\nabla F_t(w_t)\|^2 \leq F_0(w_0) - F_T(w_T) + T \cdot O(\eta^3 L^2 \tau^3) + \sum_{t=0}^{T-1} \Delta F_t(w_{t+1}).$$

Using $\sum_{t=0}^{T-1} \Delta F_t(w_{t+1}) \leq M\Delta_{\max}$ we obtain

$$\frac{3}{4}\eta\tau \sum_{t=0}^{T-1} \|\nabla F_t(w_t)\|^2 \leq F_0(w_0) - F_T(w_T) + T \cdot O(\eta^3 L^2 \tau^3) + M\Delta_{\max}.$$

Divide by $\tfrac{3}{4}\eta\tau T$ to get the averaged-stationarity bound

$$\frac{1}{T} \sum_{t=0}^{T-1} \|\nabla F_t(w_t)\|^2 \leq \frac{C_1}{\eta\tau T} + C_2\,\eta^2 L^2 \tau^2 + \frac{C_3 M\Delta_{\max}}{\eta\tau T}, \tag{17}$$

where $C_1, C_2, C_3 > 0$ are explicit constants (traceable to the $3/4$ factor and the constants hidden in the $O(\cdot)$ term). (We have removed any $\sigma^2$ term because in the full-batch setting $\sigma^2 = 0$.)

We must respect the one-step descent requirement used earlier, in particular choose $\eta$ so that

$$\eta \leq \frac{1}{4L\tau}. \tag{S}$$

Under (S) the coefficient of $\|\nabla F_t\|^2$ in the one-step bound remains positive.

The RHS of equation 17 contains two $\eta$-dependent terms:

$$\Phi(\eta) \;=\; \frac{C_1}{\eta\tau T} + C_2\,\eta^2 L^2 \tau^2 + \frac{C_3 M\Delta_{\max}}{\eta\tau T}.$$

Combine the $1/(\eta\tau T)$ terms:

$$\widetilde{C} := C_1 + C_3 M\Delta_{\max}, \qquad \Phi(\eta) = \frac{\widetilde{C}}{\eta\tau T} + C_2\,\eta^2 L^2 \tau^2.$$

Minimizing $\Phi(\eta)$ over $\eta > 0$ (subject to (S)) gives

$$\eta^\star = \Big(\frac{\widetilde{C}}{2C_2 L^2 \tau^3 T}\Big)^{1/3} = \Theta\Big(\frac{1}{L^{2/3}\tau\, T^{1/3}}\Big).$$

Pick the largest step size allowed by (S), e.g.

$$\eta = \frac{1}{4L\tau}.$$

Plugging this constant $\eta$ into equation 17 yields

$$\frac{1}{T} \sum_{t=0}^{T-1} \|\nabla F_t(w_t)\|^2 = O\left(\frac{L}{T}\right) + O\left(\frac{\zeta^2}{\tau}\right) + O\left(\frac{M\Delta_{\max}L}{T}\right),$$

which (absorbing $L$ into constants) is the commonly used full-batch FedAvg-style rate

$$\frac{1}{T} \sum_{t=0}^{T-1} \|\nabla F_t(w_t)\|^2 = O\left(\frac{1}{T}\right) + O\left(\frac{\zeta^2}{\tau}\right) + O\left(\frac{M}{T}\right).$$

Since $M$ is a fixed finite constant, we have $O(M/T) = O(1/T)$, and thus the bound simplifies to

$$\frac{1}{T} \sum_{t=0}^{T-1} \|\nabla F_t(w_t)\|^2 = O\left(\frac{1}{T}\right) + O\left(\frac{\zeta^2}{\tau}\right).$$

### A.4 ABLATION STUDIES

#### A.4.1 SELECTION OF SCALING FACTOR

The scaling factor $s$ for the FedChill algorithm was determined through a one-dimensional sweep over values in the range $0.5$ to $3.5$, conducted across 10-, 20-, and 50-client settings on CIFAR-10 with their respective `frac` values. As shown in Table 10, the choice of $s = 2.0$ consistently yielded strong performance, achieving the best or near-best accuracy across different configurations, and was therefore selected as the final value.

#### A.4.2 SCALING FACTOR VS. HETEROGENEITY MEASURES

An ablation study was conducted to evaluate different heterogeneity measures against the scaling factor (effectively, the range of temperature initialization) Here, $s = 0.5$ and $s = 3$ were chosen as the extremities of the scaling factor range to study the impact of different heterogeneity measures on performance. As shown in Table 11, the choice of heterogeneity measure directly affects both local and server accuracies. For $s = 3$, the Jensen–Shannon (JS) (Nielsen, 2020) divergence provided the best trade-off, achieving the highest local (70.33%) and server (55.97%) accuracies. Conversely, when $s = 0.5$, the performance was more balanced across different measures: KL divergence yielded the highest local accuracy (69.85%), while Entropy Difference achieved the highest server accuracy (55.18%). These results highlight that the JS divergence is particularly effective at capturing distributional closeness in higher scaling regimes, while KL and Entropy Difference remain competitive at lower scaling values (Shlens, 2014).

Table 10: Ablation Study: Scaling Factor ($s$)

| Scaling Factor ($s$) | 10 (0.1) | 20 (0.2) | 50 (0.5) |
|---|---|---|---|
| 0.5 | 49.42% | 57.74% | 60.13% |
| 1.0 | 49.81% | 58.47% | 60.76% |
| 1.5 | 51.06% | 58.67% | 61.40% |
| 2.0 | **51.68%** | 59.38% | **62.70%** |
| 2.5 | 48.62% | **59.81%** | 61.11% |
| 3.0 | 49.39% | 56.20% | 58.41% |
| 3.5 | 47.60% | 56.32% | 57.75% |

Table 11: Ablation Study: Heterogeneity Measures vs. Accuracy

| $s$ | Measure | Local (%) | Server (%) |
|---|---|---|---|
| 3 | Ratio | 69.80 | 50.98 |
| | KL | 68.16 | 47.41 |
| | JS | **70.33** | **55.97** |
| | Entropy Diff | 66.47 | 49.94 |
| 0.5 | Ratio | 68.92 | 54.89 |
| | KL | **69.85** | 54.53 |
| | JS | 68.95 | 54.66 |
| | Entropy Diff | 69.88 | **55.18** |

## A.5 CORRELATION ANALYSIS DIAGRAMS

This analysis investigates whether alternative client characteristics can better predict optimal temperature settings than the heterogeneity score used in the main FedChill implementation. We tested multiple client metrics across 10 clients with varying data distributions under a Dirichlet partitioning scheme ($\alpha = 0.5$). A similar study was also repeated using 50 clients. The evaluated characteristics included heterogeneity score, Gini coefficient, dominant class probability, number of active classes, dataset size, Shannon entropy, and KL divergence (Nielsen, 2020) (Shlens, 2014).

The correlation analysis revealed that the heterogeneity score remains the strongest single predictor of optimal temperature ($r = -0.389, p = 0.267$), supporting the hypothesis that clients with higher heterogeneity benefit from lower temperatures. Other metrics such as the Gini coefficient ($r = 0.198$) and dominant class probability ($r = -0.160$) showed only weak correlations, while multi-variable combinations did not improve predictive power. When examining performance gains, dominant class probability exhibited the strongest negative correlation ($r = -0.361$), but most other measures showed little association with improvements (Patil et al., 2022) .

The scatter plots illustrate the relationships between each predictor and optimal temperature settings. Trend lines indicate the direction and strength of correlations, with the heterogeneity score plot showing the clearest downward trend, consistent with the FedChill hypothesis that more heterogeneous clients benefit from lower temperatures. Overall, although no alternative predictor outperformed the heterogeneity score in the 10-client setting, the analysis reinforces its role as a principled basis for temperature assignment.

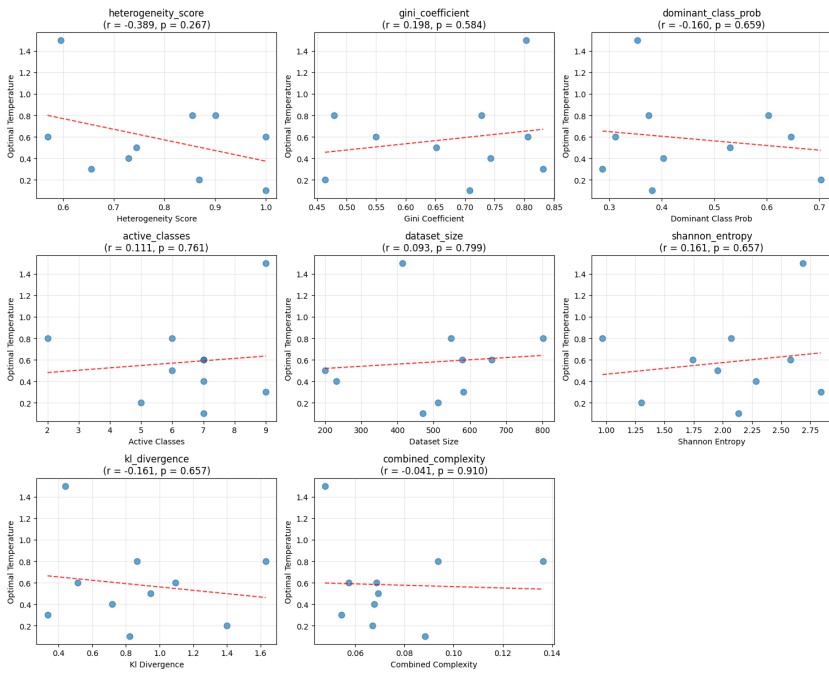

Figure 3: (10 clients) Correlation analysis between client characteristics and optimal temperature values. Scatter plots show relationships between various client metrics and optimal temperatures

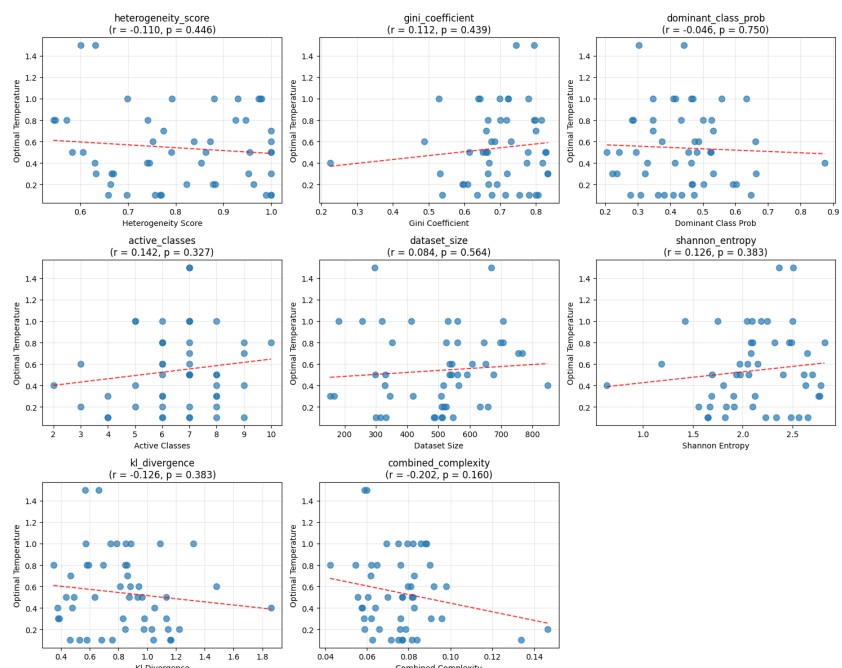

Figure 4: (50 clients) Correlation analysis between client characteristics and optimal temperature values. Scatter plots show relationships between various client metrics and optimal temperatures.

## A.6 DISCLAIMER

The convergence analysis in Appendix A.3 was developed with the assistance of a large language model and manually verified to the best of our abilities. Furthermore, large language models were used modestly to assist in polishing the writing of this paper.

