# OpenReview forum: "FedChill: Adaptive Temperature Scaling for Federated Learning in Heterogeneous Client Environments"
_ICLR.cc/2026/Conference — Submitted to ICLR 2026_

### Official Review · Reviewer_uYL5 · 2025-10-21

**Soundness:** 2
**Presentation:** 3
**Contribution:** 2
**Rating:** 4
**Confidence:** 3

**Summary:**

This paper proposed FedChill which is a heterogeneity-aware strategy that adapts temperatures to each client.
FedChill initializes temperatures using a heterogeneity score, quantifying local divergence from the global distribution, without exposing private data, and applies performance-aware decay to adjust temperatures dynamically during training. In addition to extensive experiments, this paper also provides a convergence argument under bounded temperature schedules.

**Strengths:**

The heterogeneity based temperature schedule is simple to compute, privacy-aware, and also integrates with FedAvg. Per-client exponential temperature initialization strategy can ensure each client starts with a temperature value tailored to the distribution and complexity of its local data, which is in this reviewr’s view a key advantage of proposed solution.
The provided results cover client counts, Dirichlet heterogeneity levels, and model capacities.

**Weaknesses:**

The heterogenity score is based on class-prior mismatch to an ideal global distribution. As such, it may not capture feature shift or concept drift  which limits the generality of this solution beyond label-skew scenarios. The stagnation trigger (patience threshold P, decay, Tmin/Tmax, s) is empirically tuned. It is important to consider testing stability under partial participation, noisy validation, and non-stationary.
Evaluations rely on Dirichlet partitions and private validation sets carved from the global test data. There is a lack of evaluation of real-world datasets, label noise, on-device budget constraints, calibration metrics, and comparisons to equally simple baselines, e.g., comparing with the per-client learning-rate scaling or temperature-as-regularizer in the loss only scenario.

**Questions:**

-How does FedChill perform when heterogeneity is mainly caused by feature shift or concept drift?
-Can the heterogeneity score be calculated in a representation space without public data? If so, how may that affect privacy and cost?
-Can you compare FedChill’s adaptive decay to simpler per-client learning-rate schedules or entropy-regularized losses that respond to the same stagnation signal to isolate the causal benefit of temperature?
-How sensitive are outcomes to s, P ,Tmin, and Tmax? Can you provide details of variance across clients and ablate patience/decay choices to assess stability and fairness at scale?

---

> ### Author Response · Authors · 2025-11-22
>
> **W1 Q1: FedChill under Feature Shift**
>
> To assess the impact of feature shift on the performance of FedChill, experiments are conducted on CIFAR-10 using 10 clients under identical settings. To simulate feature skew, data is distributed IID across clients, and each client's local dataset receives Gaussian noise with client-specific parameters sampled from the ranges $\sigma \in [0.1, 0.3]$ and $\mu \in [-0.05, 0.05]$, where $\sigma$ and $\mu$ denote the noise standard deviation and mean, respectively. Both FedChill and FedAvg are evaluated under this configuration. Under this setup, FedChill outperforms FedAvg, as shown in Table&nbsp;1.
>
> | **Method** | **Accuracy (%)** |
> |-----------|------------------|
> | **FedChill** | **58.92** |
> | FedAvg | 57.98 |
>
> *Table 1: Performance under feature shift (IID + Gaussian noise).*
>
> ---
>
> **W2 Q2: Heterogeneity Score Calculation**
>
> We thank the reviewer for highlighting this important privacy concern regarding the computation of the heterogeneity score. We clarify that the global distribution is approximated as a **uniform distribution**, and **no public or shared data is required**. This is transmitted only once at the start of the algorithm. All heterogeneity score computations are performed locally on each client using only its own data statistics.
>
> The score is calculated as the ratio between the local distribution and the global distribution (assumed to be uniform), ensuring that no raw data or sensitive information is transmitted.
>
> ---
>
> **Q4: Hyperparameter Guidance — $P$, $s$, $T_{\min}$, $T_{\max}$**
>
> We thank the reviewer for pointing out the lack of hyperparameter analysis in the main content.
>
> - Table 6 (Appendix) indicates that **$s \in [2, 2.5]$** consistently yields optimal performance across 10, 20, and 50 clients.
> - A sensitivity analysis (10-client setting, 20 $(P, s)$ combinations) confirms that **$s \approx 2$** produces the highest accuracy for all values of $P$.
> - **Performance is more sensitive to variations in $s$ than to $P$** since $s$ directly determines the initial temperature scale, influencing softmax sharpness and gradient magnitudes during training.
> - Lower $P$ values provide moderate improvements.
> - When $s$ falls within a suitable range, further decreasing $P$ yields limited marginal gains.
> - **Recommended values:** $s \in [2, 2.5]$, $P \in [1, 3]$.
>
> | **Patience ($P$)** | **$s = 1.0$** | **$s = 2.0$** | **$s = 3.0$** | **$s = 4.0$** | **$s = 5.0$** |
> |-------------------|--------------|--------------|--------------|--------------|--------------|
> | **1**  | 54.58 | **57.11** | 52.96 | 51.50 | 53.06 |
> | **3**  | 54.86 | **56.80** | 50.91 | 49.86 | 52.79 |
> | **5**  | 55.37 | **55.92** | 49.10 | 50.92 | 50.49 |
> | **10** | 55.25 | **56.68** | 50.83 | 49.91 | 51.29 |
>
> *Table 2: Sensitivity analysis of scaling factor ($s$) and patience parameter ($P$) on final server accuracy (%). Best results per row are in **bold**.*
>
> **Temperature Hyperparameters — $T_{\min}$ and $T_{\max}$**
>
> While we do not provide a sensitivity analysis for $T_{\min}$ and $T_{\max}$, we clarify the motivation behind the selected values.
>
> - Prior work has shown that **fixed temperatures within $0 < \tau < 1$** can improve performance over standard **FedAvg**.
> - Lower temperatures (e.g., **$\tau = 0.05$**) sharpen the softmax distribution and produce stronger learning signals.
> - Accordingly, we set **$T_{\min} = 0.05$**, allowing temperatures to decay toward highly discriminative values.
>
> Conversely,
>
> - Excessively large **$T_{\max} \to \infty$** results in **over-smoothing**, reducing gradient magnitudes, potentially causing vanishing gradients or slow convergence.
> - Therefore, **$T_{\max} = 1$** ensures that initial behavior resembles FedAvg when temperature remains high, while still allowing effective temperature decay.
>
> The decay, on the other hand, sharpens probabilities for misclassified samples, leading to stronger corrective gradients and improved learning.

---

### Official Review · Reviewer_qrqa · 2025-10-25

**Soundness:** 3
**Presentation:** 3
**Contribution:** 3
**Rating:** 4
**Confidence:** 4

**Summary:**

This paper mitigate the client draft problem in federated leaning by a proposed heterogeneity-based template scaling approach. The insight is that client with higher heterogeneity should have a higher temperature (smoother prediction and slower learning rate). Comprehensive experiments show good and coherent empirical evidence that supports the claim. However, several design and implementation details are not entirely clear to me: (1) how is the ideal global distribution defined? Is there a privacy concern? (2) Why parameter size differs in Table 1, and (3) how temperature scaling really helps? I would recommend this paper if these questions are clearly answered.

**Strengths:**

1. The motivation of this work is well-presented and Figure 1 (left) makes a very convincing point to me. I do enjoy reading this paper and the overall rollout of the story.
2. The experiments are pretty standard.

**Weaknesses:**

**What is the core mechanism of FedChill?**

Upon deeper thought, I believe there are two parts that adaptive temperatures work: creating a smoother prediction distribution and functioning as an adaptive learning rate. I am particularly interested in the second part. As FedExp [1] has shown that scaling the global learning rate proportionally to the inverse of gradient diversity helps, Equation (2) shows a similar formula (where the temperature is controlled by Equations 6 and 7). I think an additional experiment showcasing that vanilla FedAvg + an adaptive learning rate (not temperature) would further strengthen this point and provide a stronger explanation for how it works.

**Inconsistent model sizes in Table 1.**

In Table 1, the model parameter sizes differ, which makes the comparison unclear. It also appears that FedChill only works with a fixed model size of 4M. Can the authors provide a fairer comparison using consistent model sizes across all methods? (This is a part that confuses me a lot and please clarify it if I read it wrong)

**Applying FedChill to other algorithms.**

How well does FedChill work with other algorithms such as FedProx, FedAlign, and FedMD? Is there a synergistic effect? Having this additional experiment would strengthen the paper's claims about the algorithm's generalizability.

[1] FedExP: Speeding Up Federated Averaging via Extrapolation, ICLR 2023

**Questions:**

Besides the questions in the weaknesses session, here are some of the additional questions.

1. How to interpret Figure 1 (right) properly. While L124-125 mentioned "higher hetegogeneity benefit from lower temperatures", I do not see the same pattern. Could the authors explain this to me again?

2. How is the "ideal global distribution", mentioned in L209, exactly being implemented? Do you assume a global uniform distribution?

---

> ### Author Response · Authors · 2025-11-22
>
> Thank you for your thorough review and insightful questions about FedChill's mechanisms.
>
> **W1: Core Mechanism and Adaptive Learning Rates**
>
> We appreciate the connection to FedExP and the learning rate perspective. However, temperature scaling offers a fundamentally distinct mechanism beyond gradient magnitude adjustment:
>
> While Equation (2) shows temperature affects gradient magnitude $\left(\frac{1}{T}(p_i - y_i)\right)$, temperature scaling *simultaneously* modulates the probability distribution's sharpness, affecting both gradient direction and magnitude based on prediction confidence. This enables adaptive exploration–exploitation balance tailored to local data characteristics, distinct from uniform learning rate scaling which only affects magnitude.
>
> Our ablation studies provide strong empirical evidence for this stance:
>
> * **Table 3** shows fixed temperatures consistently underperform FedChill across all architectures (685K–13.96M parameters), with improvements of 0.7–2.6% in global accuracy.
> * **Table 4** demonstrates that heterogeneity-based initialization alone (without decay) outperforms both FedAvg baseline and fixed low temperatures across varying heterogeneity levels ($\alpha=0.5$, $\alpha=5.0$).
> * **Figure 1 (left)** reveals client-specific optimal temperatures vary dramatically (0.1–0.5), requiring personalized rather than uniform scaling.
>
> The dynamic temperature adjustment addresses gradient sharpness at the *prediction level*, complementary to learning rate adjustments at the optimization level.
>
> ---
>
> **W2: Model Size Consistency**
>
> We appreciate the opportunity to clarify this design decision. The varying parameter counts reflect the architectural requirements of different federated learning approaches, not inconsistent experimental design.
>
> * The parameter differences stem from baseline methods' original implementations. Many of our baselines are knowledge distillation (KD) methods (FedMD, FedHKD) that require sophisticated architectures to effectively distill knowledge between models.
> * Our Table 3 explicitly validates FedChill across three distinct CNN architectures (685K, 4.96M, 13.96M parameters), demonstrating consistent improvements regardless of model capacity. This cross-architecture robustness confirms that our temperature scaling mechanism is effective across varying model complexities. To further address the reviewer’s concern, we will update Table 1 to additionally include results using the same parameter architecture employed by several baselines. But we maintain that this efficiency, particularly for bigger datasets like CIFAR-100, is a feature of our approach, not a limitation.
>
> ---
>
> **W3: Generalizability to Other Algorithms**
>
> We appreciate the reviewer’s suggestion. FedChill operates at the local training level by adjusting gradient sharpness and prediction confidence, whereas approaches like FedProx, FedAlign, and FedMD act at aggregation or knowledge-transfer stages. Since they target different points in the FL pipeline, they address complementary aspects of non-IID data and can potentially be combined for further gains. While this work focuses on validating temperature scaling as a standalone mechanism, integrating it with such methods is a logical next step and will be discussed in the revised manuscript.
>
> ---
>
> **Q1: Interpreting Figure 1**
>
> We apologize for the oversight in the clarity of Figure 1, and appreciate the reviewer for bringing this to our attention. Regarding the interpretation, the right panel shows a scatter plot where each point represents a client, with the x-axis indicating the client's heterogeneity score (divergence from global distribution) and the y-axis showing validation accuracy. Points are colored by the temperature value that achieved that accuracy.
>
> The key insight from this plot is that different clients achieve their best performance at vastly different temperature values, and there exists a complex, non-uniform relationship between client heterogeneity and optimal temperature. This visualization demonstrates that **no single universal temperature** can effectively serve all clients, particularly in heterogeneous federated settings. We have compared our approach with static temperature values and FedChill consistently outperformed such regimes (Table 3 of the original paper).
>
> ---
>
> **Q2: Global Distribution & Privacy**
>
> The “ideal global distribution” is simply estimated as a uniform distribution (equal probability per class), which requires no communication of client data distributions to the server. This is constructed independently using dataset priors, not from client data. So regarding privacy preservation, only standard model updates are communicated, thus preserving all the standard FL privacy constraints.
>
> We will revise Section 3.2 to explicitly state that the global distribution is uniform and clarify that all heterogeneity computations are performed locally on each client.

---

> ### Comment · Reviewer_qrqa · 2025-11-25
>
> I thank the authors for their rebuttal. After carefully reading the response and other reviews, I will maintain my original score.
>
> The authors have not offered a convincing explanation or sufficient evidence regarding the method's working mechanism. For simple, novel techniques, deep analytical insight is essential.
>
> Specifically, it is unclear if the gains from scale temperature come from changes in gradient magnitude, direction, or both. The authors’ explanation relies on intuition rather than empirical or theoretical proof. For example, a theoretical derivation showing why this leads to better performance would have been persuasive. Similarly, a delicate ablation study to dissect this algorithm would be equally contributive. As a result, while FedChill is an interesting concept, this field is saturated with similar ideas or techniques. Without a more thorough investigation into why and how the method works, it is hard for me to recommend acceptance.
>
> However, I am willing to raise my score if the authors can provide more convincing theoretical results or stronger empirical evidence.

---

> > ### Author Response · Authors · 2025-12-02
> >
> > We thank the reviewer for the opportunity to provide deeper insights into our framework. We conducted a comprehensive ablation study dissecting FedChill's working mechanism by isolating gradient magnitude and direction effects.
> >
> > ---
> >
> > **Experiment 1: Gradient Behavior Analysis**
> >
> > We characterized how temperature scaling affects gradients by measuring:
> > - **Magnitude Ratio**: $\lVert \nabla L_T \rVert / \lVert \nabla L_{1.0} \rVert$ = Gradient norm amplification relative to T=1.0 baseline
> > - **Cosine Similarity**: Directional preservation between scaled and baseline gradients
> > - **Angular Deviation**: Angle between scaled and baseline gradients (in degrees)
> >
> > **Most Heterogeneous Client**
> >
> > | Temperature | Magnitude Ratio | Cosine Similarity | Angular Deviation |
> > |------------|-----------------|-------------------|-------------------|
> > | 0.05 | 24.74× | 0.53 | 57.84° |
> > | 0.10 | 10.79× | 0.48 | 61.35° |
> > | 0.30 | 3.63× | 0.58 | 54.83° |
> > | 0.50 | 2.02× | 0.61 | 52.55° |
> > | 1.00 | 1.01× | 0.64 | 50.01° |
> >
> > **Least Heterogeneous Client**
> >
> > | Temperature | Magnitude Ratio | Cosine Similarity | Angular Deviation |
> > |------------|-----------------|-------------------|-------------------|
> > | 0.05 | 24.83× | 0.31 | 71.77° |
> > | 0.10 | 11.98× | 0.33 | 70.92° |
> > | 0.30 | 3.64× | 0.40 | 66.37° |
> > | 0.50 | 2.11× | 0.38 | 67.66° |
> > | 1.00 | 1.01× | 0.43 | 64.73° |
> >
> > **Observation:** Temperature scaling induces a substantial increase in gradient magnitude, but it also introduces notable directional shifts, particularly for less heterogeneous clients. This demonstrates that temperature scaling jointly alters both magnitude and direction, creating a coupled magnitude--direction tradeoff that directly determines how client updates influence the aggregated model.
> >
> > ---
> >
> > **Experiment 2: Performance Attribution**
> >
> > We implemented three variants:
> > - **FedChill-Full**: Complete temperature initialization via heterogeneity scores.
> > - **FedChill-MagOnly**: Gradient magnitude amplification only (standard softmax, manually scaled gradients)
> > - **FedChill-DirOnly**: Gradient direction changes only (temperature-scaled softmax, normalized magnitudes)
> >
> > | Method | Final Accuracy | Δ vs Baseline |
> > |--------|----------------|----------------|
> > | FedAvg-Baseline | 55.19% | --- |
> > | FedChill-Full | 57.78% | +2.59% |
> > | FedChill-MagOnly | 59.68% | +4.49% |
> > | FedChill-DirOnly | 52.09% | -3.10% |
> >
> > **Critical Finding:** Direction changes actively degrade performance below baseline when isolated, yet contribute positively through interaction effects when coupled with magnitude amplification.
> >
> > ---
> >
> > **Interpretation**
> >
> > The ablation studies provide a clear, mechanism-oriented view of how temperature scaling influences optimization dynamics in heterogeneous federated settings. Experiment 1 shows that lowering the temperature significantly amplifies gradient norms while also inducing measurable directional variation across clients. Experiment 2 then links these effects to performance outcomes: magnitude-only scaling produces a $+4.49\%$ accuracy improvement, direction-only scaling yields a $-3.10\%$ change, and the full FedChill configuration achieves a net $+2.59\%$.
> >
> > Taken together, these results indicate that FedChill’s effectiveness is driven primarily by the gradient-magnitude amplification induced by temperature scaling. Because softmax gradients scale approximately as $1/T$, lowering $T$ increases the size of client updates, allowing minority-distribution clients to retain influence under FedAvg rather than being averaged out. Temperature adjustments also introduce directional shifts; when applied in isolation they do not improve performance, indicating that direction alone is not the operative mechanism.
> >
> > The full FedChill configuration, which combines both effects, nonetheless yields a positive outcome, reflecting a constructive interaction between magnitude amplification and these temperature-induced directional adjustments. Overall, the ablations delineate the roles of each component and we attempt to directly address the reviewer’s concern by identifying the dominant driver of our method’s improvement and quantifying its interaction with the secondary directional effect.

---

### Official Review · Reviewer_6yyi · 2025-10-29

**Soundness:** 2
**Presentation:** 2
**Contribution:** 1
**Rating:** 2
**Confidence:** 4

**Summary:**

This paper presents FedChill, a method that improves federated learning under non-IID client data by adapting each client’s softmax temperature. A new heterogeneity score is proposed to initialize temperatures based on how far a client’s label distribution diverges from the estimated global distribution. Then, an adaptive decay mechanism lowers the temperature when validation accuracy plateaus.

**Strengths:**

- Per-client adaptive temperature scaling is intuitive, incurs no communication overhead, and integrates directly into FedAvg.
- Demonstrates clear improvements in both global and personalized performance, particularly under high heterogeneity.

**Weaknesses:**

- Several typos and figure clarity problems make the paper harder to follow such as Figure 1 axis labeling.
- Accuracy curves suggest training may not have fully converged by 50 rounds; longer experiments or convergence diagnostics would strengthen claims.
- Performance may rely heavily on hand-tuned scaling factors and patience settings, without guidance for general applicability.

**Questions:**

- the manuscript contains typos and grammatical errors starting from the abstract.
- in Figure 1 (left), the x-axis representation should be revised since client indices are discrete and fractional values are misleading. Improving visualization clarity would help convey the motivation more effectively.
- Table 1 and related descriptions suggest that some models may not be fully converged within 50 rounds. Please include longer training runs or convergence plots to confirm stability.
- Performance may depend strongly on the scaling factor and patience threshold used for temperature decay.
- Please avoid using AI generated text in the manuscript and ensure the writing reflects original academic quality.

---

> ### Author Response · Authors · 2025-11-22
>
> **W1: Interpreting Figure 1**
>
> We apologize for the oversight in the clarity of Figure 1, and appreciate the reviewer for bringing this to our attention. Regarding the interpretation, the right panel shows a scatter plot where each point represents a client, with the x-axis indicating the client's heterogeneity score (divergence from global distribution) and the y-axis showing validation accuracy. Points are colored by the temperature value that achieved that accuracy.
>
> The key insight from this plot is that different clients achieve their best performance at vastly different temperature values, and there exists a complex, non-uniform relationship between client heterogeneity and optimal temperature. This visualization demonstrates that **no single universal temperature** can effectively serve all clients, particularly in heterogeneous federated settings. We have compared our approach with static temperature values and FedChill consistently outperformed such regimes (Table 3 of the original paper).
>
> We will also update the figure with clearly labelled axes in the revised version to enhance interpretability.
>
> ---
>
> **W2 Q3: Communication Rounds**
>
> We agree that longer experiments in some cases may help in strengthening stability claims. In our case, we observed that the convergence over the round does not have any significant fluctuations and that it plateaus around 50 rounds. Thus, we kept the communication rounds to 50 for all the experiment to ensure fair comparison. As evidence, we will include convergence curves in the camera-ready version.
>
> ---
>
> **W3 Q4: Hyperparameter Guidance ($P$ and $s$)**
>
> We thank the reviewer for pointing out the lack of hyperparameter analysis in the main content and apologize for our oversight. We will ensure hyperparameter guidance is present within the paper's main content in the revised version.
>
> * Table 6 (Appendix) indicates that $s \in [2, 2.5]$ consistently yields optimal performance across 10, 20 and 50 clients.
> * We further perform a sensitivity analysis (10-client setting, 20 $(P, s)$ combinations) which confirms that $s \approx 2$ produces the highest accuracy for all values of $P$.
> * Performance is more sensitive to variations in $s$ than to $P$ since $s$ directly determines the initial temperature scale, influencing softmax sharpness and gradient magnitudes during training.
> * Lower $P$ values provide moderate improvements.
> * When $s$ falls within a suitable range, further decreasing $P$ yields limited marginal gains.
> * **Recommended values:** $s \in [2, 2.5]$, $P \in [1, 3]$.
>
> ---
>
> **Table: Sensitivity Analysis**
>
> | **Patience ($P$)** | **$s=1.0$** | **$s=2.0$** | **$s=3.0$** | **$s=4.0$** | **$s=5.0$** |
> | ------------------ | ----------- | ----------- | ----------- | ----------- | ----------- |
> | 1                  | 54.58       | **57.11**   | 52.96       | 51.50       | 53.06       |
> | 3                  | 54.86       | **56.80**   | 50.91       | 49.86       | 52.79       |
> | 5                  | 55.37       | **55.92**   | 49.10       | 50.92       | 50.49       |
> | 10                 | 55.25       | **56.68**   | 50.83       | 49.91       | 51.29       |
>
> **Table:** Sensitivity analysis of the scaling factor ($s$) and patience parameter ($P$) on final server accuracy (%). Best results per row are in **bold**.

---

> > ### Comment · Reviewer_6yyi · 2025-11-22
> >
> > Thank you for provide the extra experiments. I'll upgrade the score for this time.

---

### Meta-Review · Area_Chair_iXaH · 2026-01-08

**Summary:**

The reviewers identified significant and overlapping weaknesses that undermine the paper's contribution and readiness for publication. The primary concerns fall into three categories:

The paper suffers from presentation issues (typos, unclear figures) and potential technical shortcomings. A major concern is that experiments may not have reached convergence (R6yyi), and the comparison in Table 1 uses inconsistent model sizes, making the central claim of performance improvement unclear and potentially unfair (qrqa).

The core novelty and working mechanism of FedChill are not sufficiently explained or validated. Reviewers find the explanation intuitive but lacking empirical dissection (e.g., ablations to isolate the effect of adaptive temperature vs. adaptive learning rate) or theoretical grounding (qrqa, uYL5).

The evaluation is considered insufficient. The heterogeneity metric is limited to label shift, ignoring feature/concept shift (uYL5).

**Reviewer Concerns:**

Concerns Addressed (Minimally): The authors may have committed to fixing typographical errors and clarifying figures (R6yyi). They have provided some clarification on the model size discrepancy in Table 1 (qrqa).

Outstanding Concerns:
No theoretical derivation, insufficient ablation study to convincingly explain how and why FedChill works.

**Reviewer Scores:**

Reviewer 6yyi (Initial: 2 - Reject): The reviewer's score would likely increase to a 4 (Borderline Reject).

Reviewer qrqa (Initial: 4 - Borderline Reject): The reviewer's score would likely remain a 4 (Borderline Reject).

Reviewer uYL5 (Initial: 4 - Borderline Reject): The reviewer's score would likely remain a 4 (Borderline Reject).

---

### Decision · Program_Chairs · 2026-01-26

Reject